# Formulation and Evaluation of Loperamide HCl Oro Dispersible Tablets

**DOI:** 10.3390/ph13050100

**Published:** 2020-05-18

**Authors:** Blasco Alejandro, Torrado Guillermo, Peña M Ángeles

**Affiliations:** 1Centro Militar de Farmacia de la Defensa. Carretera M-609 de Miraflores, Km 34, Colmenar Viejo, 28770 Madrid, Spain; alejandro.blasco@edu.uah.es; 2Unidad Docente de Farmacia y Tecnología Farmacéutica, Facultad de Farmacia, Universidad de Alcalá, Alcalá de Henares, E-28871 Madrid, Spain; guillermo.torrado@uah.es

**Keywords:** loperamide hydrochloride, orodispersible tablet, physicochemical characterization

## Abstract

This work proposes the design of novel oral disintegrating tablets (ODTs) of loperamide HCl with special emphasis on disintegration and dissolution studies. The main goal was augmenting the adherence to treatment of diseases which happen with diarrhea in soldiers who are exposed to diverse kinds of hostile environments. Optimized orally disintegrating tablets were prepared by the direct compression method from galenic development to the industrial scale technique, thanks to strategic and support actions between the Spanish Army Force Lab and the Department of Biomedical Sciences (UAH). The results show that loperamide HCl ODT offers a rapid beginning of action and improvement in the bioavailability of poorly absorbed drugs. The manufactured ODTs complied with the pharmacopeia guidelines regarding hardness, weight variation, thickness, friability, drug content, wetting time, percentage of water absorption, disintegration time, and in vitro dissolution profile. Drug compatibility with excipients was checked by DSC, FTIR, and SEM studies.

## 1. Introduction

CEMILFARDEF (Centro Militar de Farmacia de la Defensa) is the newest Spanish military lab placed in Colmenar Viejo (Madrid, Spain), and develops the production, supply and maintenance of health resources in agreement with the needs of the Spanish Army Forces and the Spanish National Defence [1,2]. Nowadays, Spain participates in sixteen international military missions with more than 2500 military personnel in four different continents [3]. This means that potential military patients and local people could be in diverse scenarios including hazardous situations in which a bacterial infection or dehydration with diarrhoea could be a potential health risk. According to the pharmaceutical military medicines official list, there are various kind of chemical, biological, radiological and nuclear defence (CBRN) antidotes and strategical medicines in which the loperamide hydrochloride hard tablet is included [4], and it is also in the listings in the World Health Organization Model List of Essential Medicines [5]. Loperamide HCl (4-[4-chlorophenyl]4-hydroxy-N-dimethyl-alpha,alphadiphenyl-1-piperidine-butanamide hydrochloride) is a synthetic antidiarrheal µ-opioid receptor agonist that primarily affects receptors in the intestine. Loperamide HCl is efficacious in the treatment of diarrhoeas [6,7]. Diarrhoea is by far the most common medical problem among people travelling to less developed tropical and subtropical countries. Travellers’ diarrhoea (TD) describes the symptoms of an intestinal infection caused by certain bacteria, parasites, or viruses.

Loperamide HCl systemic bioavailability is about 0.3% because of first-pass metabolism [8] and it is almost extracted and metabolized by cytochrome P450 in the liver where it is conjugated [9]. Peak plasma concentrations occur within 2.5 h of oral administration of the solution and 5 h of the gelatine capsule [8]. Loperamide HCl presents self-limiting adverse effects because of low oral absorption and the incapacity to cross the blood–brain barrier; both circumstances, together with short-term administration of the drug for most indications, explain its minimal central nervous system effects [8].

For all of the above, the main objective of this work is the development and the scale up of a galenic formulation in solid dosage forms as antidiarrheal for oral use on the management of TD, choosing orodispersible tablets (ODTs) to optimize benefits and diminish harms and problems associated with this common acute infection. The choice of ODTs is based on two different aspects: (i) the need to promote bioavailability because of the potential patient´s tactical scenario and (ii) the industrial production view. This study develops an ODT using a dose of 2 mg of loperamide HCl. The financial global crisis in the early 21st century motivated changes in the pharmaceutical industry like cost-containment measures [10], the improvement of political management and manufacturing operations [11] to still guarantee the quality implementation and harmonization [12].

ODTs are solid dosage forms that disintegrate rapidly in the oral cavity in 1 min, in the presence of saliva without any difficulty of swallowing, which provides a rapid beginning of action, and therefore a rapid absorption and high bioavailability of the drug, due to the presence of superdisintegrants. This likewise implies other important and varied advantages with respect to its stability, administration without water, precise dosing and easy manufacturing that guarantees a great adhesion to the treatment. Therefore, it can be said that a sublingual absorption and the no drinking water requirement are situations that are very vital in those places where potable water is not available, and it is a plus point compared with the urge to drink water. An ideal ODT has a pleasant mouth feel, enough hardness and acceptable friability limit and needs conventional manufacturing methods [13]. ODTs have become an alternative to other conventional solid dosage forms due to better patient compliance, are convenient for young children and elderly and mentally retarded persons or patients with swallowing difficulties.

In the first step of the design of the tablet, due to a simple subsequent industrial production, an industrial process is considered that leads to the manufacture of tablets by direct compression, since it represents the simplest and most cost-effective tablet manufacturing technique, and to achieve the disintegration in less than one minute, superdisintegrants will be selected, or, based on sugar [14]. Superdisintegrants include cross-linked cellulose derivative, carboxymethyl cellulose, sodium starch glycolate, and polyvinylpyrrolidone, which gives burst disintegration when in contact with water or salivary secretions. The main limiting factors of the direct compression method are the physical properties of the active pharmaceutical ingredients (API) and their concentration in the ODTs [15], so a careful selection of excipients, their proportion and the production method design is necessary to reduce those limiting variations and control other properties like flow, particle size distribution or good compression [16].

To finish, a high-performance liquid chromatography (HPLC) method was developed in order to study the correct mixture as a part of the production process and to study the API concentration in the orodispersible tablet.

## 2. Results and Discussion

Figure 1 summarizes the method structure of this study divided into four steps. The first operative actions of loperamide ODT (preformulation, formulation and pilot scale) were done in Biomedical Science´s Laboratory Department (University of Alcalá, Faculty of Pharmacy). Afterwards, the last operative action (industrial scale) was done in CEMILFARDEF. All supporting actions were done in Biomedical Science´s Laboratory Department.

### 2.1. Preformulation

An exhaustive market study was carried out to design an ODT loperamide tablet by the Spanish Army Force according to consideration of the material, physicochemical, dust and bulk properties and API biopharmaceutical properties. The excipients were selected according to the pharmaceutical manufacturing experience of the Spanish Army Force. To support these strategy actions, a background information and literature review were done by searching the Medicine Online Information Centre of Spanish Agency of Medicines and Medical Devices [17] and different kinds of official books [15,16,18,19,20,21,22]. After testing with different formulations, the definitive composition of tablet formulations is shown in Table 1.

### 2.2. Formulation

The excipients and their proportions were selected according to their function and due to technological aspects for a direct compression manufacturing process [19]. Therefore, the excipients selected during the formulation stage were binding agents, lubricants, solvents, non-sticks, additives and superdisintegrants. Loperamide ODT selected formulas were formula n°14 and formula n°15 included in Table 1 the proportion for a 150 mg ODT was: loperamide HCl (1.33%) mannitol (42.67%), calcium hydrogen phosphate dihyhidrate (42.67%), sodium starch glycolate (5.00%), magnesium stearate (1.00%), sodium cyclamate (1.00%), menthol (0.33%), anise extract (1.00%), HPMC (5.00%). The proportion for a 200 mg ODT was: loperamide HCl (1.00%) mannitol (42.835%), calcium hydrogen phosphate dihyhidrate (42.835%), sodium starch glycolate (5.00%), magnesium stearate (1.00%), sodium cyclamate (1.00%), menthol (0.33%), anise extract (1.00%), HPMC (5.00%). To guarantee an industrial process of direct compression, it was necessary to start with more diluent ratios such as calcium hydrogen phosphate dihydrate and mannitol. The selection and proportion of diluents were influenced by the low proportion of API (2 mg of an ODT of 150 mg, represents 1.33%; 2 mg of an ODT of 200 mg represents 1.00%), its solubility in water and technological aspects such as good flow properties, good compression properties and one more transition easy laboratory to production scale [23].

Calcium hydrogen phosphate dihydrate has good flow and compression properties, but it is an abrasive excipient [23], and so a lubricating excipient such as magnesium stearate was necessary. Mannitol, a good candidate for direct compression excipient [24,25], is a cohesive powder soluble in water and of good organoleptic appearance [18,26]. It is important to keep in mind that a form of mannitol with high relative humidity leads to a polymorphic transition. This moisture-induced transition that occurs during a wet granulation process, so, among other questions, the choice of our manufacturing process is justified [27]. Hypromellose (HPMC) was selected as a binder [28] in order to prevent technological problems like capping in the compression step. HPMC is an odourless and tasteless excipient and is soluble in cold water, forming a viscous colloidal solution [28] which could help in the disaggregation caused by other excipients in the ODT´s formulation. Sodium starch glycolate was selected as a superdisgregant [17,22,29,30] by rapid uptake of water disintegration (facilitated by the solubility in water of mannitol) followed by rapid and enormous swelling to ensure the disaggregation of the tablet in mouth. Sodium starch glycolate is a cross-lined starch that serves to greatly reduce water solubility, while allowing the excipient to swell and absorb many times its weight of water, causing the tablet to have a fast uniformity of disaggregation [18,30]. It is used in tablets prepared by the direct compression or wet granulation processes. It is very advantageous because increased tablet compression pressure does not influence disintegration time [29]. Finally, additives like sodium cyclamate, menthol and anise extract were used for a palatable ODT [18,31,32,33]. It was necessary to mask the bitterness of loperamide HCl [34] and to ensure treatment compliance.

The water absorption ratio and wetting time are important criteria in the understanding of the capacity of the disintegrants to swell in the presence of a small amount of water. Sodium starch glycolate was successfully used as a superdisintegrant for the loperamide orodispersible formulation, demonstrating an in vitro disintegration time of 9.63 s, a wetting time of < 3 s, a water absorption ratio of 99.40 ± 0.47%, and a cumulative drug release of ~90% after 5 min [16,35].

A simple manufacturing process is necessary to facilitate good manufacturing implementation like minimize cross contamination or a quality risk management implementation [20]. Technological aspects (Figure 1) such as CMA and CPP were considered to ensure the approval of the CQAs of the final product.

During the weighing process, we studied the behaviour of different components and ensured the correct weight tolerance. For the sieving process, we used a 0.8, 1.00 and 1.50 mm sieve size. Density bulk, density tapped, flowability and the angle of repose were studied to learn the behaviour of each ingredient and to understand their influence in the mixing process.

The mixing process started with the first minority proportion formula ingredients (loperamide HCl, sodium cyclamate, anise extract, menthol and sodium starch glycolate type A) with five minutes in 30 rpm conditions. After that, calcium hydrogen phosphate dihydrate was added with five minutes and 30 rpm conditions. Then, mannitol was added with five minutes and 30 rpm conditions and finally magnesium stearate was added with 3 min and 30 rpm conditions. A DSC study, an IR study and an SEM study were done as a process control. The compression process conditions where 5 pressure in the Bonal scale with 7 mm punch, and CQAs of the loperamide ODT were studied according to the Eur Ph [36] technology test.

### 2.3. Solid-State Characterization

DSC, FT-IR spectra and SEM were performed to examine the possible interactions between components in drugs and excipients in the formulations to ensure the quality, security and safety of the ODT. The combination of SEM studies with other thermal and spectroscopic techniques offers interesting opportunities for the characterization of incompatibilities between materials [37].

#### 2.3.1. Thermal Analysis

A DSC study (Appendix A) it was carried out by selecting mixtures of drug-excipients in a 1:1 ratio (w/w). The DSC equipment was calibrated using indium and high purity zinc as a reference material to standardize temperature and heat flux signals. The samples of approximately 3 mg were subjected to programmed heating under dynamic nitrogen gas purge according to previously explained conditions. First, it was necessary to understand the behaviour of each API and excipients (Appendix A).

Loperamide HCl crystallizes fewer than three different crystalline forms: an anhydrous polymorphic form I representing the stable polymorph of isometric crystals and the metastable form (melting point approximately 224 °C), an anhydrous polymorphic form II (melting point of approximately 218 °C); these are needles [38,39,40] and a tetrahydrated form, whose melting point is around 190 °C [41,42].

The DSC-thermogram of the original loperamide HCl powder selected in this study exhibits a single endothermic peak located at Tonset = 229.48 °C (ΔH^F^ = 1480.48 J/g), which indicates fusion, is a typical compound event crystalline anhydrous, in this case corresponding to polymorphic form I, followed by an endothermic decomposition process at temperatures above the melting point (Tonset = 259.65 °C). Mannitol has a broad endothermic peak corresponding to the fusion at Tonset = 165.81 °C and the area under the peak reveals an enthalpy of fusion at 286.07 J/g. Mannitol used in this investigation was its form II (Tonset = 165.81 °C), and also the compactability of this polymorph is higher compared to the other two crystalline forms (form I and form III).

New experimental conditions were used to study HPMC, and a cycle of 0 to 200 °C was designed at a heating rate of 200 °C/min, which allowed for observing that the Tg of HPMC occurs at 178 °C. The DSC-anise and DSC-sodium starch glycolate (type A) (Explotab^®^*)* exhibited a single endothermic event located at 166.29 and 165.83 °C, respectively. Magnesium stearate has several peaks at 62.57 and 92.73 °C due to the loss of surface water and close to 112 °C due to the fusion of magnesium palmitate, since in its composition stearic acid and palmitic acid appear (this impurity frequently occurs in commercial lots of magnesium stearate), followed by degradation at 178 °C. Calcium hydrogen phosphate (Emcompress^®^) has two endothermic events, one around 110 °C corresponding to the onset of the evaporation of the hydration water and another around 135 °C that can be associated with a phase transition of the crystal. Landin et al. [43] claimed that dehydration takes place in two steps and that it occurs according to particle size. Menthol emanates in two forms, L-menthol and dL-menthol, with different polymorphism α, β, γ and δ for L-menthol and polymorphs α, β γ for dL-menthol. The melting temperatures for L-menthol are 42.45, 36.85, 35.55 and 35.15 °C, and moreover for dL-menthol they are 32, 27.55 and 22.75 °C, respectively. DSC-menthol given a wide endotherm, so it was necessary to design a new heat–cold cycle, at high heating and cooling rates (100 °C/min). After the first heating cycle, the sample cooled at a high speed, which means that it cannot be completely crystallized to a temperature of −60 °C. In the second heating cycle, a glass transition was observed at approximately −27 °C followed by a fusion that begins around 30 °C.

Lastly, the literature indicates that monosodium cyclamate exists in two pseudopolymorphic forms [44], such as sodium cyclamate dihydrate and anhydrous sodium cyclamate. In this study hydrated form of sodium cyclamate presented an endothermic signal at 154.8 °C in the thermogram, with a shoulder at 55 °C, referring to the dehydration process. Dehydration of sodium cyclamate is a process that occurs in multiple steps spontaneously at room temperature followed by a process of decomposition around 190–200 °C due to a dimerization that leads to the formation of N, N’-dicyclohexylsulfamide and sodium sulfate. The interactions between the mixtures in these calorimetric studies are deduced by the appearance or disappearance of peaks, peak jumps especially in that associated with fusion and/or variations in enthalpy values it is not necessary these may be greater or lesser), interchangeably, they may occur changes in the shape of the peak [45] although it should be considered that some peak enlargements are due to a decrease in the purity or crystallinity of each component in the mixture.

Next, the results of the binary mixtures of the API with each of the excipients used are described (Figure 2) to maximize the possibility of observing and producing interaction. The curves exhibit a characteristic behaviour for each compound. Figure 2A,B represent the DSC of loperamide HCl, mannitol or magnesium stearate and their physical mixtures. The jump at lower temperatures of the endothermic event corresponding to the fusion of the API, from 229.48 to 185.73 °C and 198.75 °C, respectively, can be attributed to some solid–solid interaction or a reduction in individual purity but it does not necessarily mean incompatibility.

Figure 2C–E corresponds to the physical mixture with HPMC, anise and Explotab^®^; in all these cases, the fusion endotherm of the excipient has disappeared. This result has to be contrasted with the IR and SEM studies to reach a correct conclusion; that is, with these calorimetric results, no explanation can be definitive, one might think that HPMC degrades at 210 °C, which would produce a masking of the drug’s fusion endotherm in this physical mixture. These results have also been found for other drugs such as atovacone [46] or to the complete solubility of the drug in the excipient, which melted at a lower temperature than the drug [47].

Figure 2F,G present the results corresponding to the physical mixture with sodium cyclamate and Emcompress^®^, a change in the melting event matching to the active substance is observed, much more perceptible in the case of sodium cyclamate, for Emcompress^®^ occurs a jump to slightly lower temperatures of the endothermic event, but unlike the mannitol and magnesium stearate, the peak has a lower melting area, which clearly indicates that the crystallization water of calcium phosphate hydrogen partially dissolves the drug and the basic environment could contribute to this. This was verified by a new study with a second heating, it showed a broad melting peak corresponding to the drug changing at a lower temperature with a slight change in associated enthalpy. There are many active ingredients that are incompatible with this excipient for example famotidine [48], quinapril [49] or metronidazole [50].

To conclude, Figure 2H represents DSC of loperamide HCl, menthol and physical mixture. The slight reduction in the melting temperature of the drug can represent a physical interaction between both elements without indicating an incompatibility, because the average enthalpy value for the mixture is statistically equal to that found for loperamide HCl alone. More significant changes in enthalpy values would indicate a possible chemical incompatibility between them, which could lead to the partial or total loss of the pharmacological activity of the future medication.

#### 2.3.2. FT-IR

The use of spectroscopic methods such as FT-IR in preformulation has contributed significantly to the early prediction and characterisation of possible physical or chemical interactions between the drug and excipient and to assist in the rationalized selection of the most appropriate excipients in the design of dosage forms [37,51]. In Appendix A are represented the IR lectures of loperamide HCl and mannitol, calcium hydrogen phosphate dihydrate, sodium starch glycolate, magnesium stearate, sodium cyclamate, menthol, anise extract and HPMC, respectively. The IR spectrum of loperamide HCl (Appendix A) revealed characteristic absorption peaks like those previously published [52], ensuring the presence of certain functional groups. A very broad peak was obtained around 3200 cm^−1^, indicating the presence of an interchangeable proton stretch (-OH). At around 2900 cm^−1^, new peaks appear, indicating the presence of saturated carbons confirming the presence of the –CH group. Below 2000 cm^−1^, which is the region of the fingerprint, many characteristic peaks of different functional groups of the molecule are observed, such as the -CO (1475 cm^−1^), -CH_3_ (1386 cm^−1^), -R-Cl (1037 cm^−1^), and a characteristic area between 770 and 735 cm^−1^ for aromatic hydrocarbons. The infrared of the excipients selected in the preparation of the dispersible tablets are detailed in Appendix A and in Table 2 their main absorption peaks are summarized [47].

After making an accurate comparison of the infrared of the physical mixtures API + excipients and those obtained for the individual raw materials, it has been found that the infrared of the excipients, mannitol, sodium cyclamate and Emcompress^®^ showed the major differences with a clear enlargement of the highest region, possibly due to an overlap between the drug and excipient [53]. This region has been highlighted where such divergences appeared in the spectral characteristics with respect to each individual spectrum, to differentiate it in greater detail in Figure 3.

Table 3 summarizes the values of the most prominent peaks in the region indicated (green rectangle in Figure 3). The 3402.8, 3419.72 and 3736.9 cm^−1^ peaks of each of these three excipients have a value greater than 3235.93 cm^−1^ of loperamide HCl. One possible reason could be the formation of hydrogen bonds with the drug [54].

On the other hand, in the case of the sweetener, it can be explained by the possible existence of interaction between the -NH group of sodium cyclamate that interacts with the -CH_3_ group of loperamide HCl. From these values, it follows that it was the physical mixture with sodium cyclamate and Emcompress^®^, in the calorimetric studies, which showed more significant changes in the event of fusion of the API, and of minor importance for the mannitol, which only meant a shift to lower temperatures than fusion.

#### 2.3.3. SEM Studies

This technique consists of having an electron beam influence the sample. This bombardment of electrons causes the appearance of different signals that, captured with suitable detectors, provide us with information about the nature of the sample. In this analysis, a secondary electron signal (SE) was used that provided an image of the surface morphology of the sample and a backscattered signal (BSE) that gave a qualitative image of areas with different average atomic number. In order to ensure particle maintaining desired and physical characteristics during the compression manufacturing process, a SEM test was done. This technique also provided a qualitative assessment of size, shape, morphology, porosity, size distribution, crystal form, and consistency of powders or compressed dosage forms [55]. The information proportionated by SEM could guide us to ensure the defined ODT quality characterization.

Appendix A confirm represented SEM studies of loperamide HCl and the excipients selected in the final ODT (formulas n°14 and n°15). Appendix A shows the irregular crystals of the drug with regular flat surfaces and sharp edges [56]. Mannitol appears as orthorhombic needles when it crystallizes from alcohol, and anhydrous dibasic calcium phosphate appears as a white powder in the form of triclinic crystals. Explotab^®^ is shown as a hygroscopic powder in the form of irregular, ovoid or pear-shaped granules, size 30–100 mm, or even rounded. Magnesium stearate and sodium cyclamate are discovered as very fine powders, of a light white colour and with very irregular edges. Menthol is a powder of acicular or hexagonal crystals, in which its observation is difficult because the crystalline form can change over time due to the sublimation that takes place during the period of observation in the microscope. Appendix A shows the round shape and the smooth and homogeneous surface of the HPMC; this will undoubtedly allow excellent dispersion and will influence the drug release modifier. Finally, anise extract is revealed as a very heterogeneous powder of soft shapes and with very different sizes.

SEM studies have also been carried out with the drug–excipient physical mixtures but have not produced any revealing data. Nevertheless, SEM studies of cross-section ODT (formulas n°14 and n°15) (Figure 4) offered revealing results. Both are presented using a 50 and 200 µm resolution. In both cases, a well-compacted mixture is seen on whose surface large spherical particles corresponding to the sodium starch glycolate perfectly dispersed inside are visible [47].

### 2.4. Pilot Scale

Before the industrial scale up, 3 kg mixed formulations were done according to the technological process exposed in the formulation step. A sampling of the mix was done in the V-blender, in order to study API concentration. Four representative points (one point in the right side, one point on the left side, one point in the middle top and one point in the middle bottom) were taken and analysed by the loperamide HPLC method as a process control. Loperamide ODT final products were also analysed by the loperamide HPLC method.

### 2.5. Industrial Scale

Industrial scale 20 kg mixing was done in CEMILFARDEF. There were two different variations: in the mixing process a biconical mixer was used, and in the compression process, a different tablet pressmachine was used. The sieving process could not be done in CEMILFARDEF in order to not interrupt manufacturing industrial process. The Powder V-blender and biconical mixer, despite their physical differences, have the same rotation of their axis, so the CMAs, the physical, chemical, biological or microbiological properties of an input material to ensure the desired quality of output, of the intermediate product were not affected. The mixing process started with the first minority proportion formula ingredients (loperamide HCl, sodium cyclamate, anise extract, menthol and sodium starch glycolate type A) with 7 min and 20 rpm conditions. Subsequently, calcium hydrogen phosphate dihyhidrate was added with 7 min and 20 rpm conditions. Then, mannitol was added with 10 min and 20 rpm conditions and finally magnesium stearate was added with 3 min and 20 rpm conditions.

Three representative samples and the final product formula n°14 and n°15) were taken and analysed by the loperamide HPLC method as a process control. The DSC study, IR study and microscopy study were done as a process control. The compression process conditions were: ODT 150 mg or ODT 200 mg by means of a punch: 8 mm, pressure: 1.8, depth: 3.5 mm, speed: 35,000 ODT/h. In Figure 5A, it is shown that the accuracy is 100% as a part of the validation process of the method and in Figure 5B, the 200 mg ODT (formula n°15) lecture is shown.

### 2.6. Critical Quality Attributes of Loperamide ODT

Critical quality attributes (CQA) of loperamide ODT (samples ODT 150 mg and ODT 200 mg) are exposed in Table 4. ACQA is a physical, chemical, biological, or microbiological property that should meet the predefined requirements to ensure the desired product quality.

ODT 150 mg and ODT 200 mg had a bright white visual aspect. Both are grooved and presented a palatable taste thanks to the correct percentage of additives (0.33% menthol, 1.00% saccharine sodium and 1.00% anise extract) achieved in formula n°5 in order to guarantee treatment compliance. The 150 mg ODT presented a thin thickness of 2.431 mm because of the change in scale (from 7 mm punch to 8 mm punch), so a 200 mg ODT was made in order to compensate for the 8 mm punch in diameter in the scale up process, which decreases the thickness of the 150 mg manufactured in the pilot scale. It involves a slight change between both formulas (n°14 and n°15) without altering the manufacturing process by subtracting 0.33% loperamide HCl in formula n°14 and adding 0.165% of mannitol and 0.165% of calcium hydrogen phosphate dihyhidrate in formula n°15. This physical change is quite representative in the CQAs’ results of each formula, as can be seen in Table 4.

The content of loperamide HCl was similar in formula n°14 (110.90 mg; +0.65%) and formula n°15 (194.93 mg; −2.535%), so it could be said that changing an 8 mm punch does not critically affect the quality of the final product. Formula n° 15 presented a greater hardness (24.74 Nw) and minor loss (0.4975% mass loss) in the friability test compared to formula n°14 (23.68 Nw; 0.7879% mass loss), so it could be deduced that formula n°15 could afford much better suffering of productive processes. Following this line of argumentation, formula n°15 presented better results in the divisibility test, obtaining a 103.43 mg mass media and all their lectures between 85–115% mass media, compared to formula n°14, which presented a 79.94 mg mass media and one result (94.2 mg) out of 85–115% but between 115–125%. Finally, formula n°15 presented less water content (1.28%) than formula n°14 (1.41%).

Due to the small amount of API in each ODT, each solution was carried out in 50 mL of water; all results were close to 100%. For the same reason of the small concentration of API per unit, the reading was interpreted in the validated loperamide HPLC method instead of UV analysis. It was necessary to simulate the disaggregation in mouth to ensure the ODT behaviour. The test was performed at 37 ± 0.5 °C, 20 mL volume for six individual ODTs of formula n°14 and six individual ODTs of formula n°15 (Table 1). Also, to ensure the initial dose of an adult [17], the same study was also by adding two ODT of each formula six times. Each ODT formula n° 14 presented practically the same disaggregation time in artificial saliva as ODT formula n°15 and obtained similar lectures by trying with two units, so the amount of the two ODTs does not saturate the solution. This result could be promising in an API release worse case, like a dry-mouthed patient caused by a hot and dry environment.

## 3. Experimental

### 3.1. Materials

We obtained loperamideHCl (Brenntag Química S.A., Barcelona, Spain), calcium hydrogen phosphate dihyhidrate (Emcompress^®^, Fagron, Barcelona, Spain), starch (Guinama, Valencia, Spain), talc (Fagron), magnesium stearate (Guinama), hydroxypropyl cellulose (Klucel G, Sigma-Aldrich), hydroxypropylmethylcelulose (Guinama), croscarmellose sodium (Vivasol^®^, JRS Pharma, NY, USA), saccharin sodium (Guinama), menthol (Fagron), Aniseextract (Disproquima S.A.), hypromellose (VivaPHARM^®^, JRS Pharma), xylitol (UPSA S.A. laboratories), crospovidone (PVP, Sigma-Aldrich, Madrid, Spain), mannitol (Mannoge, EZ spray Dried, SPI Pharma, Wilmington, USA), mannitol (Fagron), sodium starch glycolate Type A (Explotab^®^, JRS Pharma) and sodium cyclamate (Guinama).

Dipotassium hydrogen phosphate trihydrate was purchased from Quimipur, acetonitrile (ACN) from Scharlau, and phosphoric acid from Sigma-Aldrich. HPLC grade water was obtained using a Millipore Direct-Q3 UV Ultrapure Water System (Watford, UK). Buffer solutions were used according to USP 42 [57] pH 9.0 buffer, pH 6.8 buffer, pH 4.0 buffer, pH 7.0 buffer, pH 10.0 buffer, and ACN 70%/ miliQ water 30%. All other materials used in the study were of European Pharmacopoeia (Eur Ph) grade [36].

#### 3.1.1. Preparation of Physical Mixtures

Physical mixtures of loperamide HCl and mannitol, magnesium stearate, HPMC, anise, Explotab^®^, sodium cyclamate, Emcompress^®^ and menthol physical mixtures, in 1:1 weight ratio were prepared by physically mixing the components thoroughly for 10 min in a mortar until a homogeneous mixture was obtained. The powder was then stored in a desiccator.

#### 3.1.2. Preparation of Artificial Saliva

According to Torrado et al. [58], artificial saliva was prepared. It was composed of sodium chloride (0.126 g/L purified water), potassium chloride (0.964 g/L purified water), potassium thiocyanide (0.189 g/L purified water) potassium phosphate monobasic (0.655 g/L purified water), urea (0.200 g/L purified water), sodium sulfate (0.763 g/L purified water), ammonium chloride (0.178 g/L purified water) and calcium chloride dyhydrate (0.228 g/L purified water).

### 3.2. High-Performance Liquid Chromatography (HPLC) Analysis

The quantities of loperamide HCl were determined using a validated HPLC assay. The conditions of the validated method are: mobile column ACE Excel 5 C18 150 × 4.6, 5 mm; mobile phase: ACN: acetic acid 1%at a flow rate of 1.2 mL/min. The column temperature was set to 25 ± 5 °C and pressure at 200bars with an injection volume of 15 µL; wavelength: 224 nm, using a Hewlett-Packard GMBH Series 1050 (Boeblingen, Germany). Our fast method with tR 1.9 min was developed in order to decrease the use of organic phase mobiles The method was shown to be selective with a calibration curve (y = 15.825 × + 45.91), r^2^ = 0.995 (n = 9). This method was performed by adding 20 µL of different concentrations of loperamide with 80 µL of mobile phase. The method had a concentration range of 2.0–60.0 µg/mL. The method was selective, with a limit of detection of 0.3 ng/mL and a limit of quantification of 1.0 ng/mL. The percentage RSD for the method precision was found to be less than 2.8%. The accuracy was in 97.56–102.01%. The proposed method is precise, accurate, selective and rapid for the determination of loperamide hydrochloride raw material and different ODTs. The development of loperamide HPLC validated method started by searching different kinds of studies elaborated by different authors [59,60,61,62,63] and finally was validated according to ICH Q2 (R1) (CPMP/ICH/381/95) for the determination of ODTs of loperamide.

### 3.3. Differential Scanning Calorimetry (DSC) Analysis

Differential scanning calorimetry (DSC) measurements were performed in triplicate with a Mettler TA 4000 DSC Star System instrument (Schwerzenbach, Switzerland). The thermograms of the original powder of the drugs and excipients were obtained using a heating rate of 10 °C/min, operating conditions withstanding temperature ranges from 20 to 280 °C and under nitrogen flow (20 mL/min). Samples were accurately weighed (3 mg) in aluminium sealed pans. The equipment was calibrated for baseline temperature with indium metal. In all of the study, the data of the onset were used instead of the melting or decomposition temperature because in that case the mass does not have an influence.

### 3.4. Fourier Transforms Infrared Spectroscopy (FT-IR) Analysis

Infrared spectra (FT-IR) were examined over the scanning range of 500–4000 cm^−1^ using a Fourier Spectrum 2000 spectrometer Perkin Elmer^®^ System 20,000 FT-IR (Shelton, USA, United States). The resolution was 1 cm^−1^. The spectra were recorded for each drug and excipients. Samples of 2 mg were mixed with 100 mg of KBr and gently ground in a mortar. The samples were analysed from disks of about 13 mm diameter prepared with KBr and compressed in a hydrostatic press at a force of 5 T for 2 min.

### 3.5. Scanning Electron Microscopy (SEM) Studies

In this study, the scanning electron microscope (SEM) analysis was performed using Zeiss DSM 950 (Germany) equipment. A secondary electron signal (SE) and backscattered signal (BSE) were used at 3 nm resolution. Before the examination, the samples were coated with gold to make them conductive of electricity.

### 3.6. Water Absorption Ratio

A portion of tissue paper folded twice was placed in a Petri dish of 6 mm diameter and containing 3 mL of water. A tablet was put on the paper and the time required for complete wetting was measured [64]. The wetted tablet was then weighed. The water absorption ratio, R, was determined using the Equation (1)
R = Wb−Wa/Wa × 100(1)
where Wa and Wb are the weight of the tablet before water absorption and after water absorption, respectively.

### 3.7. Preparation of Orally Disintegrating Tablets (ODTs)

Tablets were manufactured using a Sieve shakers CISA (Barcelona, Spain) for the sieving process, a Powder V-blender P Prat type B n° 41412 (Barcelona, Spain) for the mixing process and a manual tablet hardness testing instrument (Bonals n°337, Spain) for the compression process, which was mechanically tooled with a flat faced punch and die 7 mm in diameter. The compacts were produced by unidirectional compression using grooved and non-grooved punch. For the industrial scale, tablets were manufactured using a biconical mixer, Glatt Labortecnic, Spain) for the mixing process and a tablet press machine Kilian RTS 21 (Berlin, Germany), which was mechanically tooled with flat faced grooved punch and die 8 mm in diameter. The final selection pool of ingredients was composed of anise extract, calcium hydrogen phosphate dihyhidrate (Emcompress^®^), croscarmellose sodium, crospovidone, hydroxypropyl cellulose (Klucel G), hypromellose (HPMC), magnesium stearate, menthol, saccharin sodium, sodium cyclamate, sodium starch glycolate type A (Explotab^®^), starch, talc and xylitol. Technological process aspects were considered in order of the technological process design due to an easier industrial scale up, with special care taken in the mixed and compression steps with an analytical process control of a loperamide HPLC method (Figure 6). The selection of critical parameters has a high potential to manufacture quality, safe and effective drugs, and ensures that the manufacturing process is not rejected, with the important economic repercussions that this causes, so that QbD will be applied in this study. The concept of QbD provides scientific basis for product development, which includes the identification of the quality target product profile (QTPP), consisting of critical quality attributes (CQA), critical material attributes (CMA) and critical process parameters (CPP) using risk assessment.

### 3.8. Tablets Characterization

#### 3.8.1. Weight Variation

Twenty tablets were randomly selected from each batch and individually weighed using an electronic balance (balance Mettler Toledo AG 245, Schwerzenbach, Switzerland). The average weight of all tablets and percentage deviation from the mean value for each tablet were determined.

#### 3.8.2. Thickness

In the preformulation step, the thickness of the tablets was determined using a thickness gauge (Vernier calibrator, Kawasaky, Japan). Six ODTs from each batch (formulasn°14 and n°15) were used and the average values were calculated. The thickness of the final product was determined using a tablet testing instrument Pharmatest PTB 311 (Hainburg, Germany).

#### 3.8.3. Hardness

In the preformulation step, the hardness expressed as the force in Newton required crushing the tablets was evaluated using a manual tablet hardness testing instrument (Bonalsn°337, Spain). The hardness of the final product was evaluated using a tablet testing instrument Pharmatest PTB 311 (Hainburg, Germany).

#### 3.8.4. Diameter

The diameter of final product was evaluated using a tablet testing instrument Pharmatest PTB 311 (Hainburg, Germany).

#### 3.8.5. Friability Test

A sample of tablets representing 6.5 g was taken and carefully dedusted prior to testing. The tablets were accurately weighed and placed in the drum of the tablet friability (Pharmatest PTF E^®^, Hainburg, Germany). The drum was rotated 100 times at 25 rpm, and the tablets were removed, deducted and accurately weighed. The percentages of friability were calculated.

#### 3.8.6. Disintegration Time

The disintegration test was performed at 37 ± 0.5 °C in water for six tablets from each formulation, using a basket-rack assembly disintegration test apparatus (Disaggregation machine Turu-Grau, Spain). The average disintegration time was calculated. Another study was performed with artificial saliva in a 10 cm diameter Petri dish 20 mL volume at 37 ± 0.5 °C for six tests of individual tablets and another six tests of two tablets, each 20 mL.

#### 3.8.7. Dissolution Studies

The in vitro dissolutions of tablets were measured using the USP paddle method (Hanson Research SR8 SRII 8-Flask dissolution test station, United States). The 500 mL 0.2 M acetate buffer pH 4.7 was kept at 37 ± 0.5 °C and the rotating speed was 50 ± 2 rpm. Six tablets were processed in each dissolution experiment. Sink conditions were verified through the analysis of dissolution samples taken from a vessel, where amounts of the drug equivalent to three times those in the tablet were added. All samples were filtered through a 0.45 μm membrane filter and the dissolved drugs released was analysed by HPLC.

#### 3.8.8. Content Uniformity

The content uniformity test was determined taking ten tablets. The preparation satisfied the test when the content individual of each unit was between 85% and 115% of the average content. The preparation did not satisfy the test if more than one individual content was outside those limits or if an individual content was outside the limits of 75–125% of the average content. If an individual content was outside the limits of 85–115% but within the limits of 75–125% of the mean content, we determined the individual contents of another 20 units taken at random. The preparation satisfied the test if no more than one of the individual contents of the 30 units was outside the limits of 85–115% of the mean content and none were outside the limits of 75–125% of the medium content.

#### 3.8.9. Divisibility Study

The divisibility study consisted of weighing each half of the 30 units. ODTs satisfied the test if the individual mass of at most one fraction was outside the limits of 85–115% of the average mass. ODTs did not satisfy the test if the individual mass of more than one fraction is outside those limits or if the individual mass of a fraction is outside of the limits 75–125% of the average mass.

#### 3.8.10. Water Content

Water content was determined by three lectures using a Mettler Toledo LJ16 Moisture Analyser (Schwerzenbach, Switzerland). ODTs were pulverized in a mortar and proximately 1.5 g was taking for each lecture.

## 4. Conclusions

ODTs have solved numerous difficulties encountered within drug administration for a large part of the population, including patients without easy access to water. It is very interesting to develop new pharmaceutical ODT products; consequently, in this article, different formulations have been designed and formulas n° 14 and n° 15 have been selected, since they can meet the required quality as indicated by pharmacopoeia, with formula n° 15being the best option, among other circumstances for obtaining the best results of physical-chemical characterization. The selection of the excipients was excellent; sodium starch glycolate was successfully selected as a superdisintegrant. Both formulations showed low wetting time and a high water absorption ratio.

In order to achieve a rational development of safe and effective drugs, studies of characterization and drug-excipient compatibility are obligatory. HPLC, DSC, FT-IR and SEM analysis allowed an adequate adaptation in the different technological stages without affecting the quality target according to the philosophy of quality by design. Finally, the ODT selected has an adequate hardness, disintegration, friability, and dissolution profile.

## Figures and Tables

**Figure 1 pharmaceuticals-13-00100-f001:**
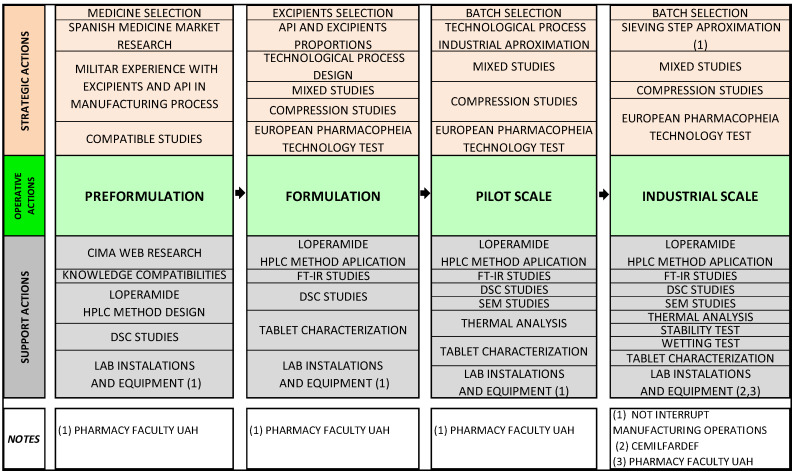
Methods structure of the study.

**Figure 2 pharmaceuticals-13-00100-f002:**
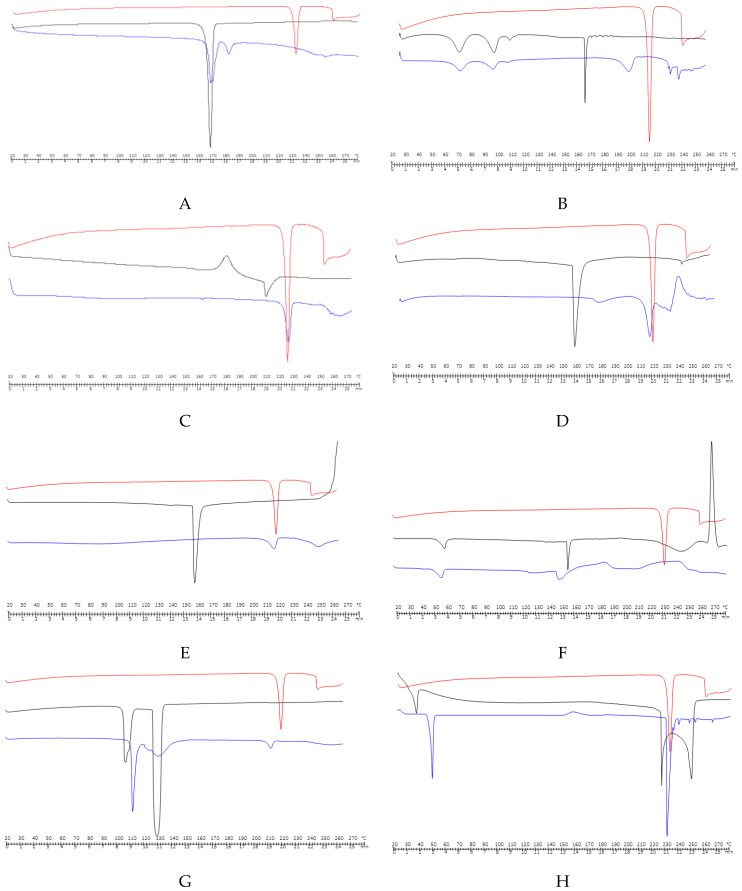
(**A**) Differential scanning calorimetry (DSC) of loperamide HCl, mannitol and physical mixture; (**B**) DSC of loperamide HCl, magnesium stearate and physical mixture; (**C**) DSC of loperamide HCl, HPMC and physical mixture; (**D**) DSC of loperamide HCl, anise and physical mixture; (**E**) DSC of loperamide HCl, Explotab^®^ and physical mixture; (**F**) DSC of loperamide HCl, sodium cyclamate and physical mixture; (**G**) DSC of loperamideHCl, Emcompress^®^ and physical mixture; (**H**) DSC of loperamide HCl, menthol and physical mixture.

**Figure 3 pharmaceuticals-13-00100-f003:**
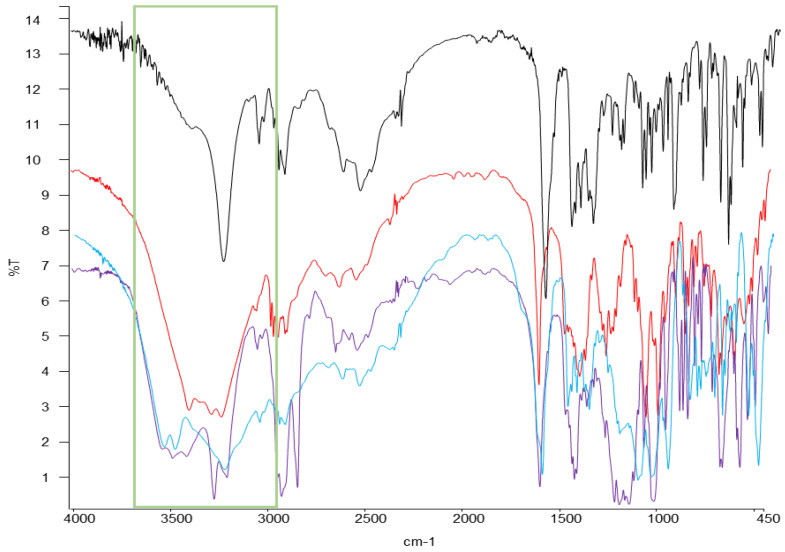
IR spectrum of the active pharmaceutical ingredient (API) loperamide HCl and excipients mannitol (red), sodium cyclamate (purple) andEmcompress^®^ (blue).

**Figure 4 pharmaceuticals-13-00100-f004:**
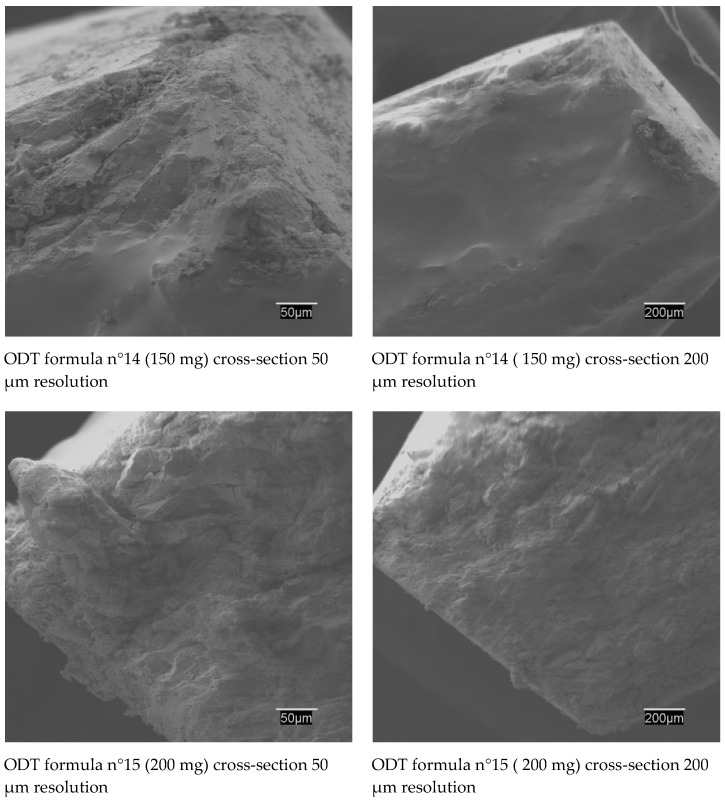
Oral disintegrating tablet (ODT) formula n°14 (150 mg) cross-section 50 µm resolution SEM image, ODT formula n°14 (50 mg) cross-section 200 µm resolution, ODT formula n°15 (200 mg) cross-section 50 µm resolution, ODT formula n° 15 (200 mg) cross-section 200 µm resolution. Change last figures 50 and 200 µm.

**Figure 5 pharmaceuticals-13-00100-f005:**
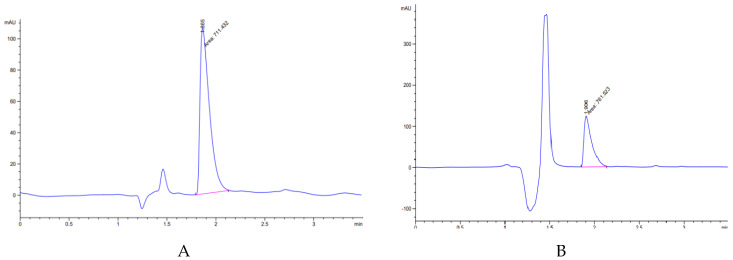
(**A**) Accuracy 100% chromatogram loperamide HPLC method, (**B**) 200 mg ODT chromatogram.

**Figure 6 pharmaceuticals-13-00100-f006:**
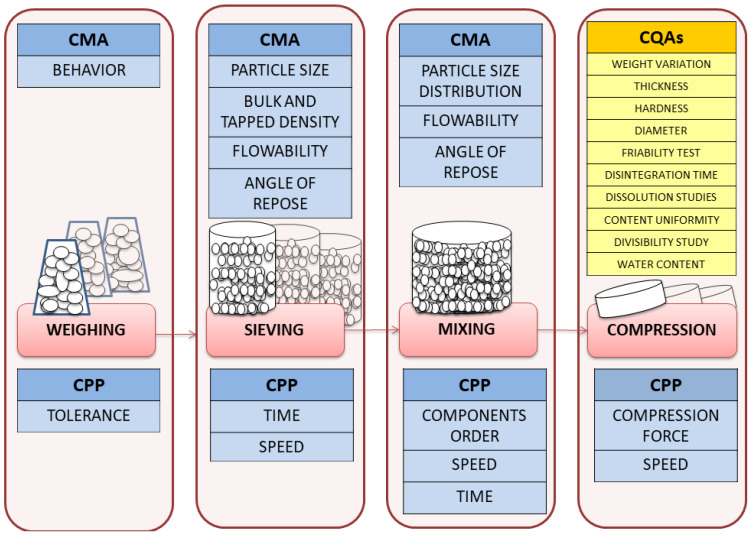
Aspects of the manufacturing process: critical material attributes (CMA), critical process parameters (CPP), product quality attributes (CQAs).

**Table 1 pharmaceuticals-13-00100-t001:** Compositions of the investigated tablet formulations.

FORMULA	LOPERAMIDE	CHPD	STARCH	TALC	MAGNESSIUM STEARATUM	HC	CS	SACCHARIN SODIUM	MENTHOL	ANISE EXTRACT	HPMC	XYLITOL	CROSPOVIDONE	MANNITOL	SODIUM STARCH GLYCOLATE TYPE A	SODIUM CYCLAMATE	OBJECTIVE	FAILURES	PHOTOS
**N°1**	1.33%	89.67%	5.00%	3.00%	1.00%	-	-	-	-	-	-	-	-	-	-	-	T	CAPPING + CHIPPING	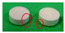
**150 mg**
**N°2**	1.33%	79.67%	5.00%	3.00%	1.00%	10.00%	-	-	-	-	-	-	-	-	-	-	T	CAPPING	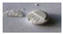
**150 mg**
**N°3**	1.33%	89.67%	-	3.00%	1.00%	-	5.00%	-	-	-	-	-	-	-	-	-	T + ES	DISGREGATION CORE´S ODT ZONE	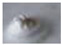
**150 mg**
**N°4**	1.33%	86.67%	-	3.00%	1.00%	-	5.00%	1.00%	1.00%	1.00%	-	-	-	-	-	-	T + P	CAP PING + P FAILURES	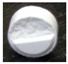
**150 mg**
**N°5**	1.33%	-	-	3.00%	1.00%	-	7.50%	1.00%	0.33%	1.00%	10.00%	74.84%	-	-	-	-	T + P	FRIABILITY TEST	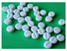
**150 mg**
**N°6**	1.33%	-	-	3.00%	1.00%	-	7.50%	1.00%	0.33%	1.00%	24.92%	49.92%	10.00%	-	-	-	T	FRIABILITY TEST	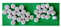
**150 mg**
**N°7**	1.33%	-	-	3.00%	1.00%	-	7.50%	1.00%	0.33%	1.00%	25.00%	30.00%	29.84%	-	-	-	T	DISGREGATION TEST	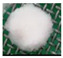
**150 mg**
**N°8**	1.33%	-	-	3.00%	1.00%	-	15.00%	1.00%	0.33%	1.00%	25.00%	22.50%	29.84%	-	-	-	T	DISGREGATION TEST	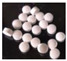
**150 mg**
**N°9**	1.33%	-	15.00%	3.00%	1.00%	-	-	1.00%	0.33%	1.00%	15.00%	33.34%	29.00%	-	-	-	T	CAPPING + CHIPPING	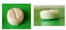
**150 mg**
**N°10**	1.33%	44.42%	-	3.00%	3.00%	-	-	1.00%	0.33%	1.00%	-	-	-	44.42%	1.50%	-	T	DISGREGATION TEST	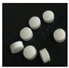
**150 mg**
**N°11**	1.33%	71.835%	-	-	10.00%	-	-	1.00%	0.33%	1.00%	-	-	-	71.835%	3.00%	-	% SSGTA	INTERFERENCE HPLC LECTURE OF SACCHARIN SODIUM	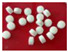
**150 mg**
**N°12**	1.33%	45.835%	-	-	1.00%	-	-	1.00%	0.33%	1.00%	-	-	-	45.835%	5.00%	-	% SSGTA	INTERFERENCE HPLC LECTURE OF SACCHARIN SODIUM	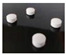
**150 mg**
**N°13**	1.33%	45.17%	-	-	1.00%	-	-	-	0.33%	1.00%	-	-	-	45.17%	5.00%	1.00%	T	CAPPING	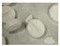
**150 mg**
**N°14**	1.33%	42.67%	-	-	1.00%	-	-	-	0.33%	1.00%	5.00%	-	-	42.67%	5.00%	1.00%	T	-	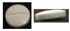
**150 mg**
**N°15**	1.00%	42.835%	-	-	1.00%	-	-	-	0.33%	1.00%	5.00%	-	-	42.835%	5.00%	1.00%	T	-	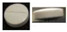
**200 mg**

**CHPD** = CALCIUM HYDROGEN PHOSPHATE DIHYHIDRATE; **CS** = CROSCARMELLOSE SODIUM; **ES** = ECONOMIC SAVINGS; **HC** = HYDROXYPROPIL CELLULOSE; **HPMC**: HYPROMELLOSE; **% SSGTA** = %SODIUM STARCH GLYCOLATE TYPE A; **T** = TECHNOLOGICAL; **P** = PALATABILITY.

**Table 2 pharmaceuticals-13-00100-t002:** Characteristic absorption peaks of the excipients used in cm^−1^.

EXCIPIENTS	OH	CH	CO	ALKYL CHAIN	CARBOXYLATE ANION	NH	SO	(H2PO4)-	COO-	AROMATIC GROUP	C-O-C
MANNITOL	3200	2947	1520								
(Figure 2**A**)
SODIUM CYCLAMATE						3400	1220				
(Figure 2**B**)
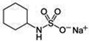
CALCIUM HYDROGEN PHOSPHATE DYHIDRATE								1040–1100			
(Figure 2**C**)
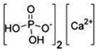
SODIUM STARCH GLYCOLATE	3270										1002
(Figure 2**D**)
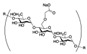
MAGNESIUM STEARATE				2916 and 2849	1446 and 1570				1567–1464		
(Figure 2**E**)
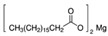
HYPROMELLOSE		1900	1400								
(Figure 2**F**)
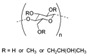
MENTHOL	3256.24			2872.22							
(Figure 2**G**)
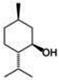
ANISE EXTRACT				2346.36						2930.29	
(Figure 2**H**)
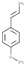

**Table 3 pharmaceuticals-13-00100-t003:** Peaks (cm^−1^) of loperamideHCl, mannitol, sodium cyclamate and Emcompress^®^.

Loperamide	Mannitol	Sodium Cyclamate	Emcompress^®^
-	3402.8	3419.72	3736.90
3235.93	3286.08	3278.31	3275.76
2959.11	3059.19	2935.42	-
2635.49	2637.2	2854.09	-
2497.04	-	2366.01	2368.92

**Table 4 pharmaceuticals-13-00100-t004:** LoperamideODT CQAs.

SAMPLES	ORGANOLEPTIC AND PHYSICAL ATTRIBUTES, DIMENSION, THICKNESS	WEIGHT VARIATION	HARDNESS	FRIABILITY	DISSOLUTION	DISGREGATION	DISGREGATION IN VITRO	WATER CONTENT	DIVISIBILITY TEST	CONTENT UNIFORMITY
**ODT** **150 mg**	Bright white Grooved Palatable Ø = 8.026 mm T = 2.431 mm *(1)*	$\bar{x}$ = 150.98 mg	23.68 Nw	W0 = 6.9047 g Wf = 6.8503 g D = 0.7879% *(2)*	nearly 100%	Between 9.00–9.57 s	$\bar{x}$ = 14.21 s *(3)*	W→ $\bar{x}$ = 1.568 g H→ $\bar{x}$ = 1.41% *(5)*	$\bar{x}$ = 79.94 mg One fractional mass out of 85–115% (94.2 mg) *(6)*	$\bar{x}$ = 2.05 mg/ODT *(7)*
							$\bar{x}$ = 13.92 s *(4)*			
**ODT** **200 mg**	Bright white Grooved Palatable Ø = 8.027mm T = 3.029 mm *(1)*	$\bar{x}$ = 194.93 mg	24.74 Nw	W0 = 6.8135 g Wf = 6.78 g D = 0.4975% *(2)*	nearly 100%	Between 9.30–9.63 s	$\bar{x}$ = 13.99 s *(3)*	W→ $\bar{x}$ = 1.519 g H→ $\bar{x}$ = 1.28% *(5)*	$\bar{x}$ = 103.43 mg All fractional mass between 85–115% *(6)*	$\bar{x}$ = 2.07 mg/ODT *(7)*
							$\bar{x}$ = 13.83 s *(4)*			

(1) Diameter (Ø) medium of 10 units. Thickness (T) medium of 10 units. (2) W0 = initial weight; Wf = final weight; D = deviation. (3) Average weight ($\bar{x}$) of six units (single tablet) in 37 ± 0.5 °C, 20 mL artificial saliva. (4) Average weight ($\bar{x}$) of twelve units (double table first Adult dose) in 37 ± 0.5 °C, 20 mL artificial saliva. (5) W→Average weight ($\bar{x}$) of three tests; H→ Average ($\bar{x}$) humidity of three tests. (6) Average weight ($\bar{x}$) of thirty units fraction mass. (7) average ($\bar{x}$) of ten units expressed in mg/ODT.

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
