# Peer review of "Formulation and Evaluation of Loperamide HCl Oro Dispersible Tablets"

_pharmaceuticals, 2020, doi:10.3390/ph13050100_

Round 1
Reviewer 1 Report
Please see the attached peer review report for this manuscript.

Reviewer 2 Report
The manuscript sound interesting. But, needs to improve related to loperamide ODT. Abstract • What is the purpose of using azithromycin and also HPLC method of azithromycin. Delete these statements. • Include results and conclusions of the investigation. Abstract not seems to e precise. It only dealt with rationale and methods. Add results section. Manuscript • the need to promote bioavailability because of the potential patient´s tactical scenario. But, there is no evidence from present investigation. • Delete Azithro HPLC section from manuscript. • HPLC - What is the composition of mobile phase for loperamide, AUFS. • Weight variation – specify the pharmacopeia which is followed. • What is pharmatest PTB 311. Describe the procedure for evaluation of thickness, hardness with this equipment. • Why distilled water selected for disintegration testing. • What is the meaning of ‘solid tablets’ in dissolution testing. • Why 500 mL volume of release media use. • What is the difference between friability testing and content uniformity in the present investigation. • Water content testing method very unclear.Author Response
- Needs to improve related to loperamide ODT Abstract
The abstract has been improved focusing on loperamide ODT.
- What is the purpose of using azithromycin and also HPLC method of azithromycin. Delete these statements.
All parts for azithromycin and its HPLC method have been removed.
- Include results and conclusions of the investigation. Abstract not seems to e precise. It only dealt with rationale and methods. Add results section. Manuscript
As explained above, the abstract has been redrafted, including research results and conclusions.
- the need to promote bioavailability because of the potential patient´s tactical scenario. But, there is no evidence from present investigation.
There is scientific evidence in the military field. The situations that a combatant faces can generate a psychological and physical response that can lead to diarrhea (Li et al., Aliment. Pharmacol. Ther., 2013; 37:799–809) and all this within a context of an austerity where not all possible sanitary means are always available and where the probability of suffering from traveler's diarrhea is higher, as can be seen in the study by Sanders where military efficiency is altered by the incidence of diarrhea in the Middle East. (Sanders et al., Curr. Opin. Gastroenterol., 2004; 21:9–14., 2004). In fact, diarrhea continues to be one of the most important threats to health to be taken into account in the military population deployed both in conflicts and in peacekeeping operations (Connor et al., Cur. Opin. Infect Diseas, 2012; 25(5):546-554) and that is that the combatant can cause hammering due to dehydration and electrolytic loss.
- Delete Azithro HPLC section from manuscript.
This part has been removed as suggested by this reviewer.
- HPLC - What is the composition of mobile phase for loperamide, AUFS.
The conditions of validated method are: mobile column ACE Excel 5 C18 150x4.6, 5 mm; mobile phase: ACN:acetic acid 1% at a flow rate of 1.2 mL/min. The column temperature was set to 25± 5 and pressure at 200 bars with an injection volume of 15 µL; wavelength: 224 nm, using a Hewlett-Packard GMBH Series 1050 (Germany).
- Weight variation – specify the pharmacopeia which is followed.
European Pharmacopoeia (Ph. Eur.) 10th Edition. 2020.
- What is pharmatest PTB 311. Describe the procedure for evaluation of thickness, hardness with this equipment.
Manual tablet testing instrument pharmatest PTB 311, is an instrument to characterize tablets (information for thickness, diameter or length and hardness). It is a dual force mode hardness test apparatus as it can be used for either linear force or linear speed increase while tablet hardness is tested.
- Why distilled water selected for disintegration testing.
We have used purified water.
We have removed the word distilled from the sections 2.8.6. and 3.6. We appreciate this correction.
- What is the meaning of ‘solid tablets’ in dissolution testing.
This term has been replaced by ODTs or tablets. We appreciate this correction.
- Why 500 mL volume of release media use.
It has been done following the criteria of the American Pharmacopoeia.
http://ftp.uspbpep.com/v29240/usp29nf24s0_m45850.html
- What is the difference between friability testing and content uniformity in the present investigation.
We apologize for the error the section content uniformity has been redrafted.
- Water content testing method very unclear.
We have used for the determination of moisture content a gravimetric method. We have improved the writing of this section.

Reviewer 3 Report
This article reports the development of loperamide HCl orodispersible tablets, from the preformulation, formulation, and pilot-scale manufacture to tablets performance assessment. In my opinion, the significance of this work has not been articulated well in the article. The discussion is superficial and not well summarized in the current version. The current version of manuscript is hard to follow. I would highly recommend revising the manuscript extensively.
Author Response
To highlight the importance of the work, very essential parts have been rewritten following the recommendations of the three reviewers. Many aspects of this work have been improved, which has helped to highlight its purpose, i.e. increase the adherence to treatment of diseases which happen with diarrhoea in soldiers proposing the design of novel oral disintegrating tablets (ODTs) of loperamide HCl.

Round 2
Reviewer 1 Report
Thank you to the authors for your responses to this reviewer's queries.
This version of the manuscript is a marked improvement over the prior version and provides a uniform story for the proposed project.
Only comment may be Figure 6 seems to be scaled oddly (the A graph is smaller than the B graph). THis coudl be due to how it is presented to us (i.e. reviewers), but the Journal and Editorial Office should also be able to help correct this action in any final proofs.
Reviewer 2 Report
The manuscript modified by the authors as per the suggestions.
Reviewer 3 Report
The revision makes this work easier to follow and understand.
I would recommend accepting this work in the current version.